# Abnormal Intrinsic Functional Hubs in Corneal Ulcer: Evidence from a Voxel-Wise Degree Centrality Analysis

**DOI:** 10.3390/jcm11061478

**Published:** 2022-03-08

**Authors:** Min-Jie Chen, Rong Huang, Rong-Bin Liang, Yi-Cong Pan, Hui-Ye Shu, Xu-Lin Liao, San-Hua Xu, Ping Ying, Min Kang, Li-Juan Zhang, Qian-Min Ge, Yi Shao

**Affiliations:** Department of Ophthalmology, The First Affiliated Hospital of Nanchang University, Jiangxi Center of National Ocular Disease Clinical Research Center, Nanchang 330006, China; 1476053262@qq.com (M.-J.C.); huangrong807@163.com (R.H.); burbo986@163.com (R.-B.L.); pyccell2009@163.com (Y.-C.P.); iamshy0626@163.com (H.-Y.S.); liaoxulin@link.cuhk.edu.cn (X.-L.L.); xusanhua6196@163.com (S.-H.X.); yypp4562021@163.com (P.Y.); km991342@163.com (M.K.); zhanglj0551@163.com (L.-J.Z.); 411438819008@email.ncu.edu.cn (Q.-M.G.)

**Keywords:** corneal ulcer, resting state, abnormal intrinsic functional hubs, degree centrality, voxel-wise

## Abstract

Background: Numerous anterior neuroimaging researches have revealed that corneal ulcers (CU) are related to changes in cerebral anatomic structure and functional area. Nonetheless, functional characteristics of the brain’s network organization still show no definite research results. The study was designed to confirm CU-associated spatial centrality distribution functional network of the whole cerebrum and explore the mechanism through which the larvaceous changed the intrinsic functional hubs. Material and Methods: In this study, 40 patients with CU and 40 normal controls (matched in sex, age, and education level) were enrolled in this study to undergo resting-state functional magnetic resonance imaging (fMRI) scans. The differences between the groups were determined by measuring the voxel-wise degree centrality (DC) throughout the whole cerebrum. For the purpose of assessing the correlation between abnormal DC value and clinical variables, the Linear correlation analysis was used. Results: Compared with normal controls (NCs), CU patients revealed high DC values in the frontal lobe, precuneus, inferior parietal lobule, posterior cingulate, occipital lobe, and temporal lobe in the brain functional connectivity maps throughout the brain. The intergroup differences also had high similarity on account of different thresholds. In addition, DC values were positively related to the duration of CU in the left middle frontal gyrus. Conclusions: The experimental results revealed that patients with CU showed spatially unnatural intrinsic functional hubs whether DC values increased or decreased. This brings us to a new level of comprehending the functional features of CU and may offer useful information to make us obtain a clear understanding of the dysfunction of CU.

## 1. Introduction

Corneal ulcer (CU), including infectious and noninfectious CU, is a common ophthalmological disease worldwide, especially in developing countries, and is the main reason of corneal opacities as well as the fourth commonest reason of monocular blindness [1]. Common causes of corneal ulcer include microbial infection, which often leads to corneal purulence [2]. (Figure 1) Bacterial keratitis, if not treated promptly, can lead to endophthalmitis or even complete loss of vision [3]. Corneal scarring may also occur after treatment for corneal ulcers. Slit lamp and fluorescein staining are common methods for the diagnosis of CU [4,5], while genomic technology is also gradually applied for corneal ulcer and other ophthalmic diseases [6,7,8]. However, both methods focus on the anterior segment and overlook other areas also involved in visual system, especially ignoring the optic tract and visual cortex.

Furthermore, the resting-state functional magnetic resonance imaging (fMRI) is a non-invasive neuroimaging mechanism that is always used to further infer the state of neural activity in specific cerebral areas by analyzing the level of blood oxygen metabolism. fMRI can detect the abnormality of cerebrum and enhances our understanding of ophthalmological diseases. There is considerable neuroimaging evidence from fMRI studies showing that ophthalmic patients may have neural tissue injury or cortical dysfunction [9].

Voxel-wise degree centrality (DC) can demonstrate our network structure at the voxel level of functional connection in the cerebrum of human beings. DC provides data regarding functional connectivity within the human cerebrum network instead of low-frequency fluctuation in regional homogeneity and amplitude [10]. Thus far, DC method has often been used for the investigation of the neuropathologic mechanisms of numerous diseases, including Parkinson’s disease, obesity, and autism [11,12,13]. Similarly, the DC method has been used to study many eye diseases so far, such as in [14].

In our research, we examined the centrality spatial distribution of the functional network throughput the whole brain in patients with CU and tried to determine the relativity between intrinsic functional hubs and clinical characteristics and which was aimed to more accurately diagnose the disease severity of patients with corneal ulcer, delay the disease progression, and guide clinical treatment more effectively.

## 2. Material and Methods

### 2.1. Participants

Our study confirmed to the ethical standards of Declaration of Helsinki as well as ethical standards of medical ethics. Relevant research methods were performed under the permission of the committee of the medical ethics of our hospital (hidden for peer review) in our research. Participants enrolled in this research are voluntary and provided written informed consent.

In this research, totally 40 CU patients (26 males, 14 females) were included. Relevant inclusion criteria of the CU group were (1) significant degeneration and necrosis in the cornea detected through slit lamp examination (Figure 1) and (2) absence of other bilateral ocular diseases (e.g., cataracts, glaucoma, etc.).

The exclusion criteria were (1) high tendency of perforation, corneal perforating, and blindness; (2) ocular injury; (3) history of systemic diseases, just like cardiopathy; (4) psychosis; and (5) other factors that interfere with DC measurement. 

Forty normal controls (NCs) (26 males, 14 females) were recruited in the study. In addition, in terms of sex, age, and level of education, they are precisely matched with the CU patients. NCs reached the following standards: (1) without neurological disease or psychosis (e.g., Parkinson’s disease, depression, bipolar affective disorder, etc.) and (2) can be scanned with MRI (e.g., no metal products in the body, such as pacemakers, etc.).

### 2.2. MRI Parameters

We performed the MRI scanning by applying a 3-Tesla MR scanner (Trio, Siemens, Munich, Germany).The T1-weighted images of the whole brain were collected through a spoiled gradient-recalled echo sequence with appropriate parameters (i.e., repetition time: 1900 ms; echo time: 2.26 ms; thickness: 1.0 mm; gap: 0.5 mm; acquisition matrix: 256 × 256; field of view: 250 × 250 mm; flip angle: 9°). Functional images were collected with appropriate parameters (i.e., repetition time: 2000 ms; echo time: 30 ms; thickness: 4.0 mm; gap: 1.2 mm; acquisition matrix: 64 × 64; flip angle: 90°; field of view: 220 × 220 mm; 29 axial).

### 2.3. fMRI Data Processing

All data were pre-filtered with MRIcro (www.MRIcro.com, accessed on 5 January 2022) and pretreated through Statistical Parametric Mapping (SPM8, http://www.fil.ion.ucl.ac.uk/spm, accessed on 5 January 2022), Data Processing Assistant for Resting-State fMRI (DPARSFA, http://rfmri.org/DPARSF, accessed on 5 January 2022), and the Resting-state Data Analysis Toolkit (REST, http://www.restfmri.net, accessed on 5 January 2022). Deleting the first 10 time points, we then obtained the residual 230 volumes. Volumes with x, y, or z directions >2 were not taken into account. In a previous study [15], a detailed description of this method was provided.

### 2.4. DC

On the basis of the individual voxel-wise functional network, our research try to calculate DC through calculating the amount of meaningful suprathreshold correlations (or the degree of the binarized adjacency matrix) between participants. In our research, all the voxel-wise DC maps were transferred into z-score maps with the application of the following equation: Zi = DCi meanall/stdall (Zi = the z score of its voxel; DCi = the DC value of its voxel; meanall = the mean DC value of all voxels in the brain mask; stdall = the standard deviation of the DC values of all voxels in the brain mask).

### 2.5. Statistical Analysis

In order to carry out demographic and clinical measurements, the SPSS version 19.0 software (IBM, Armonk, NY, USA) was utilized in the purpose of differentiating clinical features between the CU and NCs by performing, unaided, two-sample *t*-tests.

The spatial centrality distribution (hubs) throughout human cerebrum functional network was identified through one sample *t*-test for voxel-wise DC. Furthermore, two-sample *t*-tests were independently conducted, considering age, sex, and education level as interference variable in this research within the default gray matter mask. The main purpose of this operation was to assess the differences between groups in the voxel-wise DC through the application of the REST V1.8. (*p* < 0.05 for multiple comparisons with the use of Gaussian Random Field theory, z > 2.3, and cluster-wise corrected *p* < 0.05 denoted statistical significance).

Correlation analysis was applied for the purpose of evaluating the relativity of the average DC values and behavioral characteristics.

## 3. Results

### 3.1. Demographic Information and Visual Measurements

Between the two groups (*p* > 0.05), we can see from the Table 1 that there were no apparent differences either in weight (*p* = 0.824) or age (*p* = 0.892). Additionally, the average standard deviation of the duration of CU was 8.10 ± 3.57 days (Table 1).

### 3.2. Differences in DC

The distributions of the functional centers (high degree centrality) of CU patients were highly similar with NCs in spatial distribution and mainly located in the frontal lobe, precuneus, inferior parietal lobule, posterior cingulate, occipital lobe, and temporal lobe (Figure 2 and Figure 3 (red) and Table 2). Intergroup differences, on the basis of the different correlation thresholds (r_0_ = 0.15, 0.2, 0.25, 0.3, and 0.35), were also largely similar (Figure 4). Hence, we mostly reported the value of DC when the correlation thresholds were set at the standards of 0.25 in the weighted graph. In DC of two groups, Figure 5A shows the average changes.

### 3.3. Correlation Analysis

We can see a positive correlation between DC values and the duration of CU in the left middle frontal gyrus (r = 0.849, *p* < 0.001) in the DC group (Figure 5B).

### 3.4. Receiver Operating Characteristic (ROC) Curve

It is speculated that DC can be used as significant biomarker in order to distinguish the CU group from the NC group. In different cerebrum areas, we utilized ROC curve in the purpose of evaluating the average DC values. The area under the ROC curve (AUC) indicated a diagnostic efficiency. AUC values 0.5–0.7, 0.7–0.9, and >0.9 indicated low, medium, and high accuracy, separately. The AUC values of DC in specific areas were showed as follows: left cerebellum posterior lobe (0.822), right inferior parietal lobule (0.788), left middle frontal gyrus (0.878), left precuneus (0.840), and left parietal lobe (0.842) (Figure 5C). 

## 4. Discussion

As far as we know, in patients with CU, the method we used in this study was the first to be utilized to figure out the synchronous neural activity changes. However, prior to this study, DC had been successfully used in many eye diseases (Figure 6) [15,16]. Patients with CU showed significant higher DC values in the left cerebellum posterior lobe, left middle frontal gyrus, left precuneus, left superior parietal lobule, and right inferior parietal lobule in our research. We confirmed that patients with CU and NCs could be distinguished via ROC curves.

Traditionally, relevant function of the cerebellum has been associated with movement coordination, and that the cerebellum posterior lobe controls the precise movement of eyeball is proven by some present studies [17]. Early studies in monkeys have shown that part of the cerebellum is related to the saccade and conjugates movements of the eyeball [18]. In addition, the cerebellum is connected to multiple visual areas [19], and eye movement and visual formation may have a close connection with the cerebellum posterior lobe. Patients with CU suffer from severe damage to the ocular surface, and eye movement can cause severe pain. Therefore, the activity of the cerebellum posterior lobe shows a significant increase in inhibiting eye movement. However, with the advancement and use of neuroimaging technology, we have deepened our understanding of the mechanisms of the brain, especially the cerebellum posterior lobe, regarding emotional processing [20]. Previous studies using positron emission tomography have found that patients with social anxiety have a greater possibility to show abnormal signals in the brain, manifesting as elevated cerebral blood flow. The investigators of these studies concluded that the cerebellum is related to anxiety [21]. The findings of this study are similar to those of previous studies. We performed fMRI scans on the patients prior to the operation, and the patients were generally nervous. Therefore, we believe that the high DC value was caused by the anxiety of patients and pain in the left cerebellum lobe.

Zhang found that the inferior parietal lobule, which is the kernel hub of the default mode network (DMN), exhibited a lower DC value in participants with higher negative coping scores [22]. The DMN is related to negative behavior, such as refusal to seek assistance [23]. Obviously, our patients were actively seeking treatment and showed an increased DC value in the inferior parietal lobule. Similarly, a higher DC value was also shown in the precuneus (another key node of the DMN). In the medial parietal lobe, is where the anatomical position of the precuneus is located, and it has been associated with pain perception in humans [24]. Studies have shown that it works prominently in motor coordination and working memory [25,26]. We speculated that the pain caused by CU was responsible for the abnormal DC value observed in the left precuneus in the present research. An active network is located in the precuneus, taking part in maintaining the homeostasis of the central nervous system in the resting state [27]. The DMN consists of the medial temporal lobes, including the medial frontal cortex, inferior parietal cortex, posterior cingulate cortices, and anterior cingulate cortices. The brain regions with abnormal DC values in our experiment largely overlapped with the DMN. Thus, the possibility that CU causes DMN dysfunction cannot be ruled out. 

From a different viewpoint, the inferior parietal lobule is involved spatial visual information processing related to the supramarginal gyrus, angular gyrus, and intraparietal sulcus [28]. Additionally, it is involved in various cognitive processes, such as visual-motion transformation [29], fine motion [30], and visual word recognition [31]. We concluded that CU caused corneal opacity, impairing the brain function from obtaining clear visual information. This resulted in dysfunction of the inferior parietal lobule, and the increased compensatory activity led to the recorded increase in DC value. Considering the involvement of the inferior parietal lobule in multiple cognitive processes, further research is warranted to demonstrate which of these processes lead to an increase in DC value.

The middle frontal gyrus is one of the well-studied gyri. It is part of the dorsal lateral prefrontal cortex, which is recognized by the medical community as being related to human cognitive and attention function [32]. At present, it is generally thought that cognitive impairment is a distinguishing feature of depression and pre-depression [33]. According to the Tear Film and Ocular Surface Society Dry Eye Workshop II, dry eye syndrome is often accompanied by depression [34]. Patients with CU suffer more than those with dry eye syndrome. Hence, it is reasonable to suggest that CU can cause depression in these patients, possibly explaining the observed increase in DC value in the middle frontal gyrus. (Figure 7) After further experimentation, we certainly proved that in the left middle frontal gyrus, the DC values was in proportion to the duration of CU to some extent. The findings indicate to some extents that the degree of depression in patients with CU is related to the course of disease. Collectively, the findings show that accessorial DC of the left prefrontal cortex is essential to maintain an appropriate level of cognitive and attention performance.

The superior parietal lobule, which is located near the occipital lobule and a part of the visual pathway [35], is associated with body position [36]. Anterior researches have demonstrated that the frontal lobe is responsible for the integration and coordination of visual-motor [37], while for the superior parietal lobule, its function is to transmit visual information to the frontal lobe [38]. Meanwhile, Segal et al. found that the functional connectivity between the left VI area and bilateral superior lobules was decreased for early blindness [39]. Our experiments have a clear finding that in the left superior parietal lobule, the DC value was increased in CU patients. We hypothesized that CU may impair visual-motor integration and coordination. From the conclusion of our discussion above, we have made such a summary table (Table 3) [40,41,42,43,44]. 

The ROC curve analysis can be used to distinguish patients from NCs. In this analysis, the specificity and sensitivity of each encephalic region with changes in DC value were >70%. The results revealed that the AUCs of the left cerebellum posterior lobe, right inferior parietal lobule, left middle frontal gyrus, left precuneus, and left parietal lobe were 0.822, 0.788, 0.878, 0.840, and 0.842, respectively (all >0.7). This indicates that the DC value in these regions can serve as a reliable marker to distinguish patients with CU from NCs and is important for the exploration of neural mechanisms and the psychological state of patients with CU. 

The experiment still has many limitations, such as the small sample size and lack of cover and the fact that the experimental limit is not very comprehensive (such as the influence of race). At the same time, further experimental investigation is also not enough (such as infectious and non-infectious ulcer brain activity variations). In future study, we will try to improve the deficiency of the experiment.

In conclusion, the results of this study proved that CU patients showed changes in synchronic neural activity in different cerebral regions, including visual-motor-related and pain-related brain regions. These alterations can be used for the exploration of neural mechanisms in patients with CU.

## Figures and Tables

**Figure 1 jcm-11-01478-f001:**
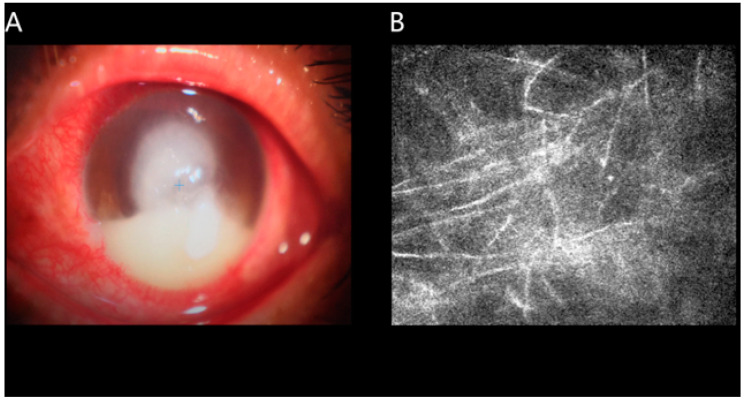
An example of UC detected with the use of slit lamp examination. Corneal infiltration was observed under the slit lamp. (**A**,**B**) shows confocal corneal microscopy of fungal keratitis.

**Figure 2 jcm-11-01478-f002:**
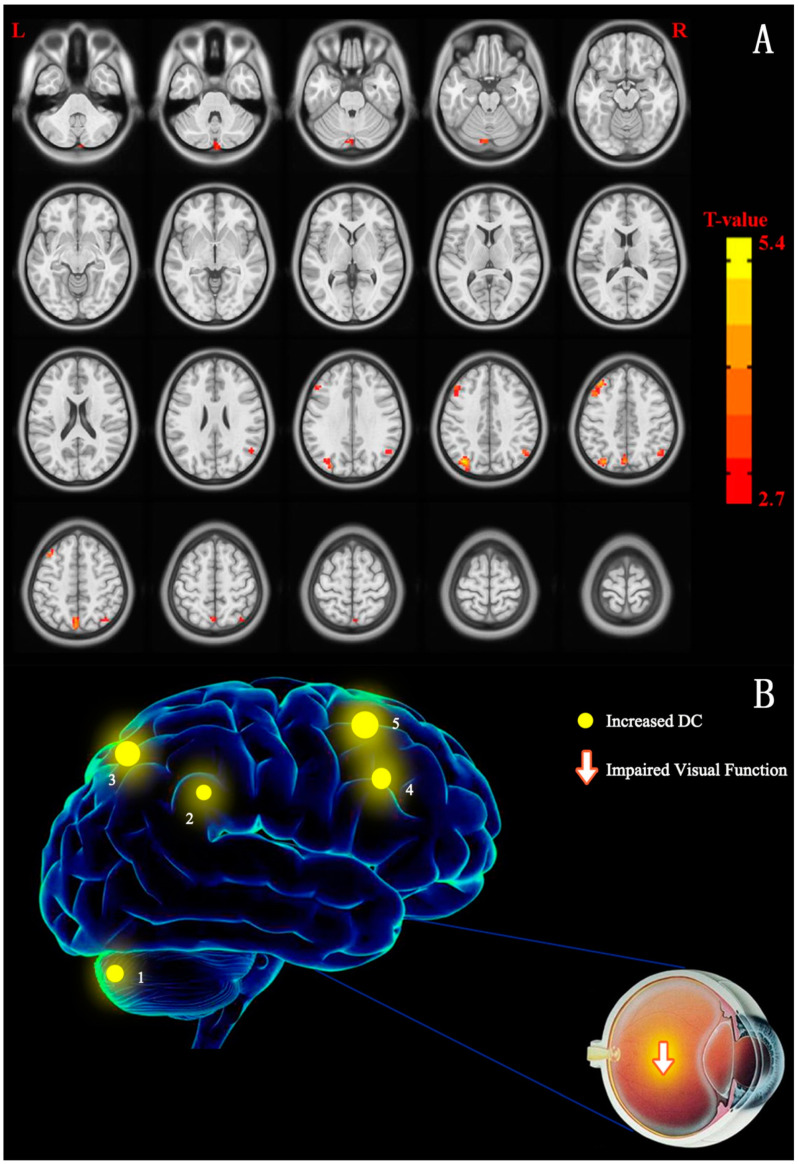
(**A**) Cross-section view: voxel-wise comparison of DC in CU and NC groups. DC values that have significant differences were observed in the frontal lobe, precuneus, inferior parietal lobule, posterior cingulate, occipital lobe, and temporal lobe. In the picture, the red and blue indicate increased and decreased DC values, respectively. Only increased DC values were recorded (*p* < 0.01, cluster > 40 voxels, AlphaSim corrected). (**B**) The DC values of cerebrum activity in CU group. In comparison with NCs, the DC values that have significant differences in these cerebrum areas were decreased to different degree in patients with CU: 1-cerebellum posterior lobe (*t* = 3.604), 2-superior parietal lobule (*t* = 4.975), 3-inferior parietal lobule (*t* = 3.526), 4-precuneus (*t* = 4.746), and 5-middle frontal gyrus (*t* = 4.239). Notes: The size of spots is used in the study to reflected the extents of quantitative changes. Abbreviations: CU, corneal ulcer; NCs, normal controls; DC, degree centrality.

**Figure 3 jcm-11-01478-f003:**
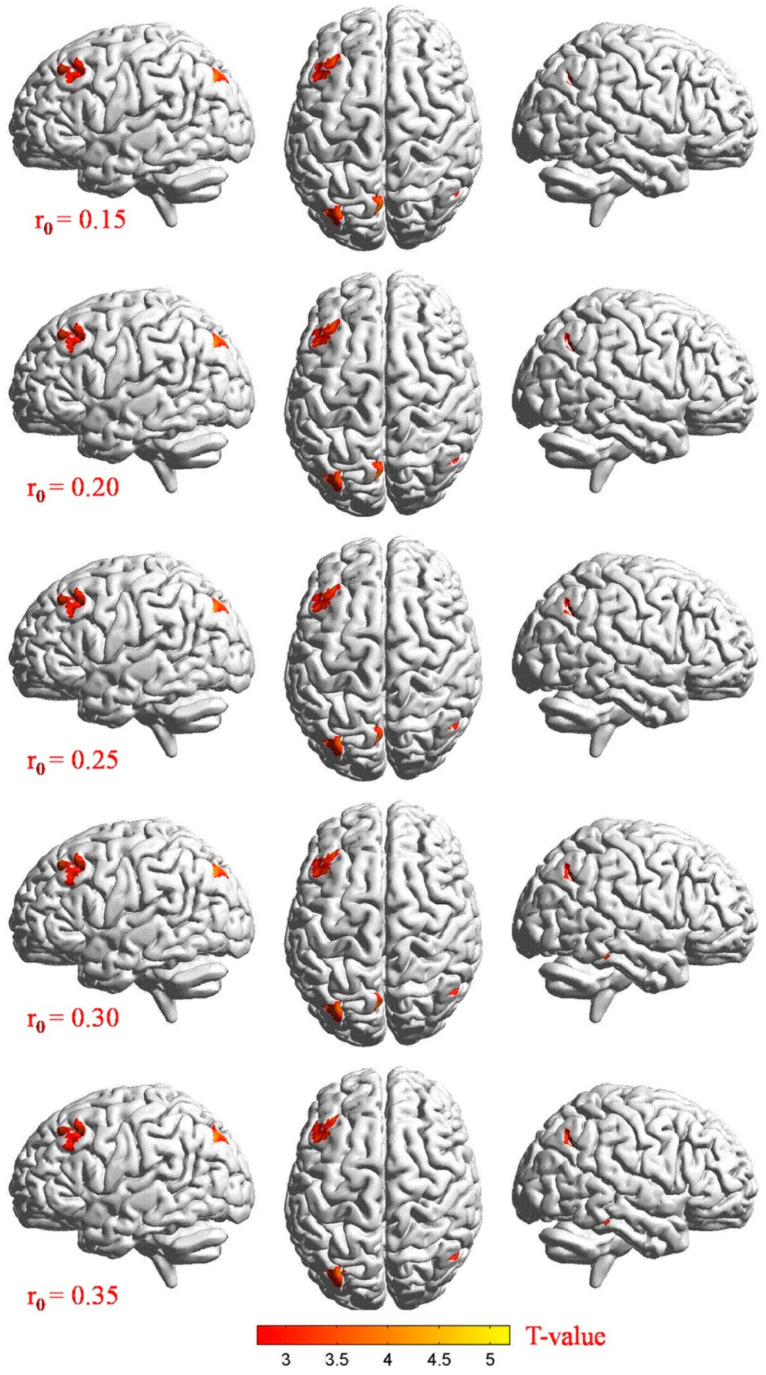
Stereoscopic view: voxel-wise comparison of DC values in two groups on the basis of different correlation thresholds (r_0_ = 0.15, 0.2, 0.25, 0.3, and 0.35). In the frontal lobe, precuneus, inferior parietal lobule, posterior cingulate, occipital lobe, and temporal lobe, we detected significant differences in DC values. Increased and decreased DC values were clearly indicated through red and blue, respectively (*p* < 0.01, cluster > 40 voxels, AlphaSim corrected). Abbreviations: CU, corneal ulcer; NCs, normal controls; DC, degree centrality.

**Figure 4 jcm-11-01478-f004:**
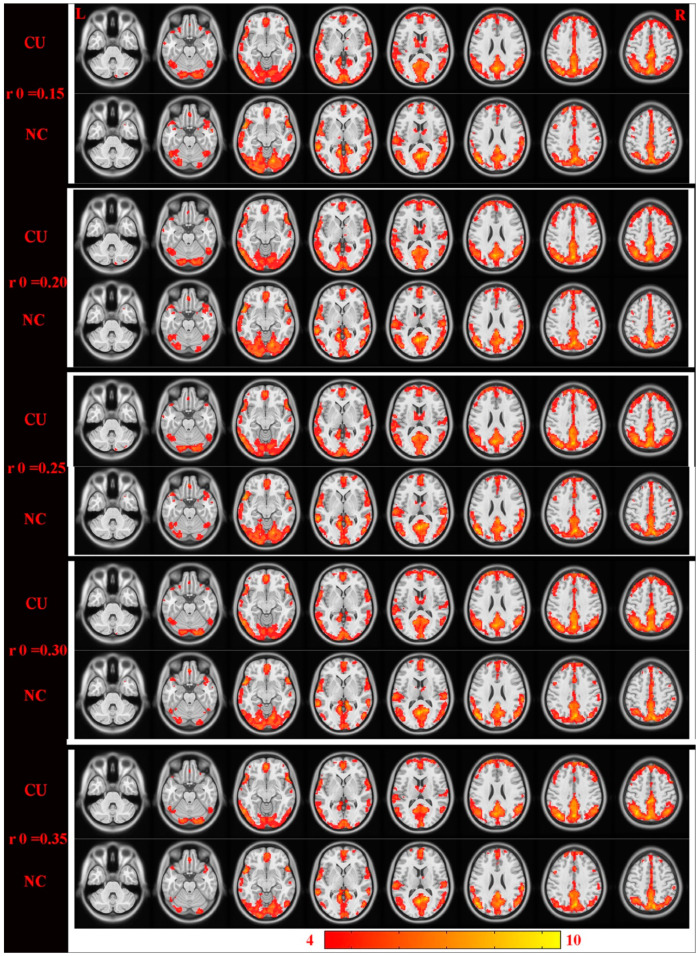
Cross-section view: intergroup differences were highly similar on the basis of different correlation thresholds (r_0_ = 0.15, 0.2, 0.25, 0.3, and 0.35). DC values that have significant differences were detected in the frontal lobe, precuneus, inferior parietal lobule, posterior cingulate, occipital lobe, and temporal lobe. Increased and decreased DC values were clearly indicated through red and blue, respectively (*p* < 0.01, cluster > 40 voxels, AlphaSim corrected). Abbreviations: CU, corneal ulcer; NCs, normal controls; DC, degree centrality.

**Figure 5 jcm-11-01478-f005:**
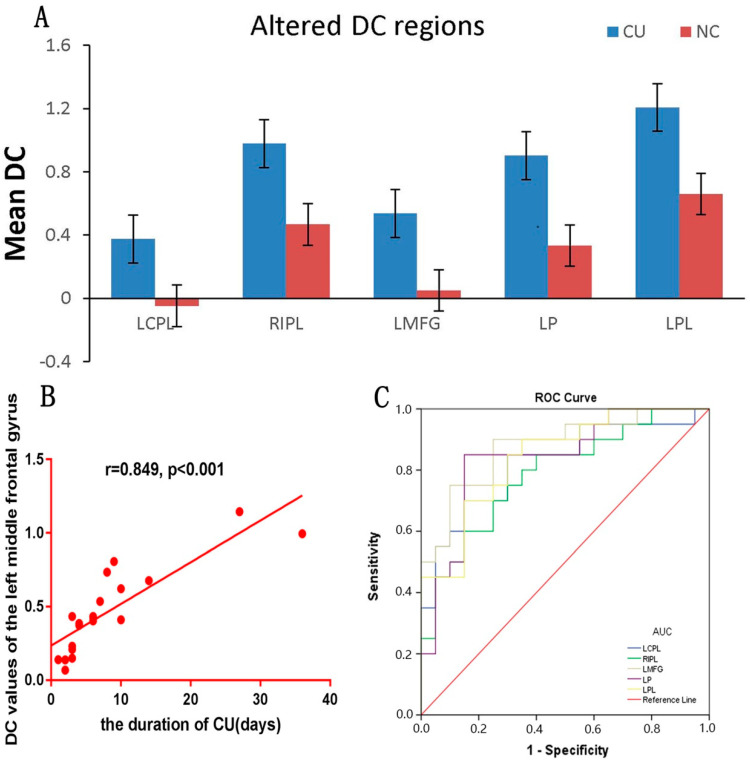
DC values of altered cerebrum regions (**A**) The average values of altered DC between two groups. (**B**) Correlations between the DC values of the left middle frontal gyrus and clinical performance. The DC values positively correlated with the duration of CU in the left middle frontal gyrus (r = 0.849, *p* < 0.001). (**C**) ROC curve analysis of the mean DC values for altered cerebrum regions. The AUCs were: LCPL, 0.822 (*p* < 0.001; 95% CI: 0.690–0.955); RIPL, 0.788 (*p* = 0.002; 95% CI: 0.647–0.928); LMFG, 0.878 (*p* < 0.001; 95% CI: 0.770–0.985); LP, 0.840 (*p* < 0.001; 95% CI: 0.713–0.967); and LPL, 0.842 (*p* < 0.001; 95% CI: 0.723–0.962). Abbreviations: DC, degree centrality; CU, corneal ulcer; NCs, normal controls; ROC, receiver operating characteristic; LCPL, left cerebellum posterior lobe; RIPL, right inferior parietal lobule; LPL, left parietal lobe, CI, confidence interval; LMFG, left middle frontal gyrus; LP, left precuneus.

**Figure 6 jcm-11-01478-f006:**
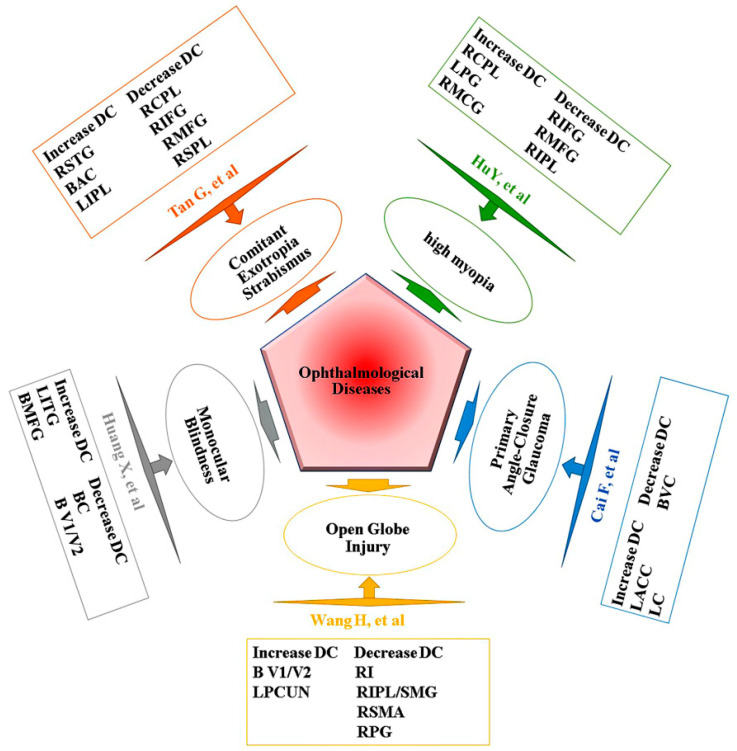
Previous studies using DC analysis. Abbreviations: DC, degree centrality; SMA, supplementary motor area; CPL, cerebellum posterior lobe; IFG, inferior frontal gyrus; SPL, superior parietal lobule; AC, anterior cingulate; IPL, inferior parietal lobule; PG, precentral gyrus; MCG, middle cingulate gyrus; LC, left caudate; VC, visual cortices; V1/V2, visual cortex I/II; MFG, middle frontal gyrus; PCUN, precuneus; RI, right insula; SMG, supramarginal gyrus; PG, postcentral gyrus; BC, bilateral cuneus; ACC, anterior cingulate cortex; STG, superior temporal gyrus; ITG, inferior temporal gyrus; L, left; R, right; B, bilateral.

**Figure 7 jcm-11-01478-f007:**
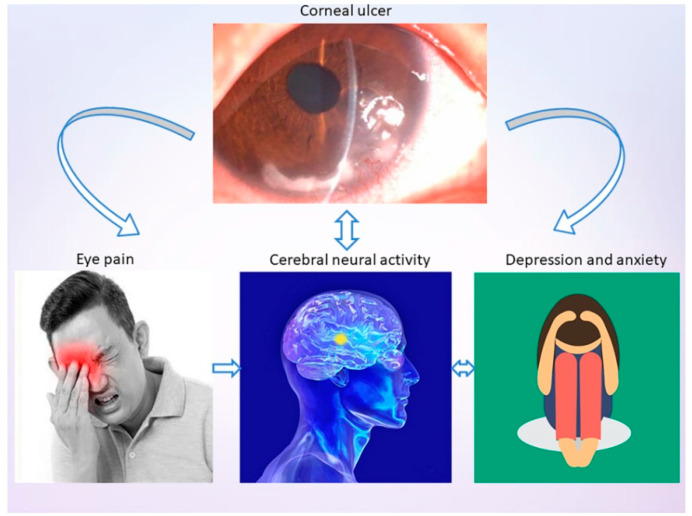
A schematic diagram of the relationship between corneal ulcer and eye pain and anxiety.

**Table 1 jcm-11-01478-t001:** The conditions of participants included in the study.

Condition	CUs	NCs	T	*p*-Value *
Male/female	26/14	26/14	N/A	>0.99
Age (years)	51.25 ± 5.46	51.98 ± 5.18	0.251	0.824
Weight (kg)	63.12 ± 7.35	63.89 ± 6.73	0.181	0.892
Handedness	40R	40R	N/A	>0.99
Duration of CU (days)	8.10 ± 3.57	N/A	N/A	N/A

* *p* < 0.05, independent *t*-tests comparing two groups. Abbreviations: NC, normal controls; N/A, not applicable; CU, corneal ulcer.

**Table 2 jcm-11-01478-t002:** Significant differences in DC between groups (r_0_ = 0.25).

Condition	L/R	Brain Regions	MNI Coordinates			Cluster Size	*t*-Value
			X	Y	Z		
CU > NC	L	Cerebellum Posterior Lobe	−9	−84	−24	49	3.604
CU > NC	R	Inferior Parietal Lobule	48	−63	42	72	3.526
CU > NC	L	Middle Frontal Gyrus	−36	36	45	104	4.239
CU > NC	L	Precuneus	−33	−75	42	93	4.746
CU > NC	L	superior parietal lobule	−3	−75	48	61	4.975

Notes: *p* < 0.01, Cluster > 40 voxels, AlphaSim corrected. Abbreviations: NC, normal control; CU, corneal ulcer; MNI, Montreal Neurological Institute; L, left; R, right.

**Table 3 jcm-11-01478-t003:** Brain regions alternation and its potential impact.

Brain Regions	Experimental Result	Brain Function	Anticipated Results
left cerebellum posterior lobe	CU > NC	sensorimotor, affective, and cognitive information [36]	Impairing both sensory and motor function; depression and anxiety [37]
right inferior parietal lobule	CU > NC	identify cognitive function, proprioception, and selective auditory attention [38]	Negative behavior and cognitive dysfunction.
left middle frontal gyrus	CU > NC	inhibition control, working memory, and emotional regulation [39]	Depression and anxiety
left precuneus	CU > NC	associated with pain perception and emotional control [40]	depression and anxiety
left superior parietal lobule	CU > NC	multiple cognitive processes and transmit visual information	impair the visual-motor integration and coordination

Abbreviations: CU, corneal ulcer; NC, normal control.

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
