# Peer review of "Abnormal Intrinsic Functional Hubs in Corneal Ulcer: Evidence from a Voxel-Wise Degree Centrality Analysis"

_jcm, 2022, doi:10.3390/jcm11061478_

Round 1
Reviewer 1 Report
I am very grateful to be able to read and review such a great work.
The authors described a comprehensive and informative work focused on possible usage functional magnetic resonance imaging(fMRI) scans in patients with corneal ulcers and matched controls. Authors cover an interesting subject that is a part in the corneal service but I still have some concerns about its clinical and practical value. The text is concise and can be read with ease. Only some major revisions are needed:
- I admit that after reading the introduction I do not fill the necessity to conduct such a research. Authors described the corneal condition and modalities that will enable the imaging process but not connection between them. No scientifical “gap of knowledge” has been specificized. Conducting the research because we can is fine but I doubt if this could lead to some practice changing outcomes. The introduction should be re-written as to encourage the significance of the study, its clinical or scientifical soundness and if possible describe the “gap of knowledge” this study is willing to fulfil. In this last case the up to date previous research should be described.
- I did not found any outcome measures or other defined measures. Please provide the primary and secondary endpoints or outcome measures. Preferably clearly stated as they can be easily reviewed and found by readers. In the present state I have the difficulties to find that.
- I am aware that this is a comparison study concerning the corneal ulcer that is a rather polymorphous in the outcomes as well as in the presentations and as such it will be difficult to gather subject for a coherent research sample. However, the sample size analysis should be performed if the previous similar research data are available or research describing the closes entitles can be used for this purpose.
- I have some concerns bout the control group – do the authors wish to prove that the corneal ulcer is a devastating disease or do they intend to show some more specific changes. I do not quite understand why healthy controls were chosen and not the treated corneal ulcers or opaque cornea due to other reasons or the participants with amblyopia. Or why the positive control (e.g. with amblyopia) and negative control (e.g. healthy participants) was not chosen? Please provide some more defence about your choice preferably with some examples or scientific based.
- Figures 2, 3 and 4 – please provide the larger figures (or better quality) it is difficult to see which part of the brain did you indicated or was brighter during the fMRI. In the figure 2A I have the impression that the fMRIs scans were stretched vertically a bit. In figure 3 you described the correlation threshold localisation within the brain and the highest value was r = 0.35 this is quite weak correlation. Could you comment how your results could be compared with other studies? Also in figure 3 as well as in figure 4 you mention red and blue regions – from this figure I cannot see any blue regions. Please provide the figures or modify the present ones to actually show the regions you mention in the caption.
- The usage of ROC in this work is difficult to understand – did the authors intention was to propose the diagnosis of corneal ulcer based on the fMRI results? Or maybe the this is the other way around the possible changes in fMRI could be associated or due to corneal ulcer? But in the second case the research group should be larger by far – especially if we take into account different described by authors localisation in the brain.
- Figure 5 again the ROC diagram is a bit small – it is difficult in the magnification of 100% to distinguish each line and their description in the 5c subsection of the figure.
In the discussion the authors mentioned a significant clinically problem – the depression that patients with corneal ulcer often suffers from. In my opinion this could be the main axis on which whole article should be spin around: to defined the possible background of depression of other mood deficits in patients suffering from corneal ulcer. The interesting question will be to know if there is some direct connection between visual loss due to corneal ulcer and moods deficits.
Reviewer 2 Report
The authors wrote an interesting paper.
Please review the following improvements:
Introduction:
Corneal ulcer (CU), including infectious and noninfectious CU, is a common oph- thalmological disease, the main cause of corneal opacities, and the fourth commonest cause of monocular blindness worldwide, especially in developing countries (1). Common causes of corneal ulcer include microbial infection, which often leads to corneal purulence (2). (Figure 1) Bacterial keratitis, if not treated promptly, can lead to endophthalmitis or even complete loss of vision (3).
I'd suggest to add the following citation for diagnosis of corneal ulcer: PMID: 35036659 PMID: 31312608 PMID: 31276030
Please review the English level
Please expand the discussion and add more limitations of the study
Background: Numerous anterior neuroimaging researches have revealed that cor- neal ulcers (CU) are related to changes in brain anatomic structure and functional area. Nonetheless, the functional characteristics of brain’s network organization are still unclear. The present study was aimed to confirm the CU-associated spacial centrality distribution functional network of the whole brain and explore the mechanism through which the larvaceous changed the intrinsic func- tional hubs. Material and Methods: In this study, 40 patients with CU and 40 normal controls (matched in sex, age, and education level) underwent resting-state functional magnetic resonance imaging(fMRI) scans. The differences between the groups were determined by measuring the voxel- wise degree centrality (DC) throughout the whole brain. Using the Linear correlation analysis to evaluate the correlation between abnormal DC value and clinical variables. Results: Compared with normal controls, CU patients showed high DC values in the frontal lobe, precuneus, inferior parietal lobule, posterior cingulate, occipital lobe, and temporal lobe in the brain functional connectivity maps throughout the brain. The intergroup differences also had high similarity on account of dif- ferent thresholds. In addition, DC values were positively related to the duration of CU in the left middle frontal gyrus. Conclusions: We found that patients with CU showed spatially abnormal intrinsic functional hubs whether the DC values increased or decreased. This brings us to a new level of comprehending the functional features of CU and may offer useful information to help us understand the dysfunction of CU.
Author Response
Thank you very much for your valuable comments. Relevant citations have been inserted into the text (see citation 6-8 for details).
The necessary paper language and discussion have been modified
Reviewer 3 Report
I have read with great interest the manuscript titled: Abnormal Intrinsic Functional Hubs in Corneal Ulcer: Evidence from a Voxel-Wise Degree Centrality Analysis, this is really a very interesting and novel work from which I have truly learned. In this study authors provide us a new knowledge about the relationship between the brain and the behavior of corneal ulcers. I would like to thank the authors for their efforts.
Some changes:
Introduction: fix fMRI. The first time that authors used need to explain the meaning (they do later in the same paragraph)
Methods: what authors want to say with ‘’ocular injury’’
I would like the authors to differentiate between infectious and non-infectious ulcers and see if they find any differences
I agree that CU patients could have changes in synchronic neural activity in many areas of the brain and the ROC curve analysis can be used to distinguish patients from NCs. However, authors need to explain (or at least hypothesize) why this is important and open new future lines of research. I really don't think fMRI is necessary to detect a corneal ulcer. However, it could be useful in chronic-cicatricial phases to see the long-term response of these patients or even in less advanced stages of a disease. It would also be interesting to correlate it with the depression reported by the authors that patients suffer
I think it is necessary to show the limitations of the study
Round 2
Reviewer 2 Report
The authors improved the paper